# Study on the Coordination of New Urbanization and Water Ecological Civilization and Its Driving Factors: Evidence from the Yangtze River Economic Belt, China

Daxue Kan [1,*], Wenqing Yao [2], Xia Liu [3], Lianju Lyu [1] and Weichiao Huang [4]

1   School of Economics and Trade, Nanchang Institute of Technology, Nanchang 330099, China;
    2011994310@nit.edu.cn
2   Business Administration College, Nanchang Institute of Technology, Nanchang 330099, China;
    2015994547@nit.edu.cn
3   School of Education, South China Normal University, Guangzhou 510631, China; 2021020820@m.scnu.edu.cn
4   Department of Economics, Western Michigan University, Kalamazoo, MI 49008, USA; huang@wmich.edu
*   Correspondence: 2011994292@nit.edu.cn

**Abstract:** For sustainable development of the world, it is crucial to solve the problems related to water environment pollution, water shortage, and the inefficient utilization of water resources during the process of urbanization in developing countries. At present, scholars mainly focus on the measurement of new urbanization (NU) and the water ecological civilization (WEC) level and the coordination relationship between NU and ecological civilization. However, there have been few studies on the coordination relationship between NU and WEC and its driving factors. We take the Yangtze River Economic Belt (YREB) in China as a case study, construct the indicator system of NU and WEC, analyze the current situation of NU and WEC in the YREB, and study the coordination state of NU and WEC in the YREB from 2011 to 2020 by using a state coordination function. We further examine the factors driving the coordination of NU and WEC by employing a two-way fixed-effects model. The results show the following: (1) The growth rate of NU and WEC in the YREB shows a fluctuating upward trend, where there is significant heterogeneity between the upper reaches, the middle reaches, and the lower reaches of the YREB. (2) The static coordination degree of NU and WEC in the YREB shows a trend of fluctuating upwards and then falling, and the dynamic coordination degree deviated from the coordinated development trajectory from 2018 to 2020. The classification of the static coordination degree of various regions in the YREB gradually becomes obvious with significant spatial aggregation characteristics, and the dynamic coordination degree of various regions has significant heterogeneity. (3) The opening-up degree, foreign direct investment, population growth, and urban–rural income gap are not advantageous to the coordination degree, while the marketization level, industrial structure, and human capital are advantageous to the coordination degree, but the regression coefficients of the latter two are not significant. The regional regression results show that the impacts of driving factors on the coordination degree have obvious heterogeneity. The research results provide a new idea and method that can be used by developing countries similar to the YERB to control water pollution, improve the ecological environment, alleviate water shortages, and improve the level of WEC in the process of NU.

**Keywords:** new urbanization; the Yangtze River economic belt; coordination degree

## 1. Introduction

Since the beginning of the 21st century, developing countries have begun to imitate developed countries in terms of urbanization in order to enhance their international status and influence. In the past two decades, developing countries made significant progress on the economic scale, foreign trade, infrastructure, and other aspects. However, they have neglected the construction of WEC for a long time (WEC is one of the important components

of ecological civilization, which means that humans use the concept of harmony between humans and water to achieve the sustainable utilization of water resources and, therefore, the sustainable development of humans. WEC advocates the harmonious coexistence of man and nature. The core idea of WEC is "harmony". Water resource conservation is the top priority of WEC construction. Water ecological protection is the key to WEC construction. WEC construction, together with economic construction and social development, is an important guarantor of sustainable development) [1,2], leading to problems, such as a prominent contradiction between the supply of and demand for water resources, the low utilization efficiency of water resources, severe pollution of water environments, and the destruction of the water ecological balance caused by the rapid accumulation of the urban population during the process of urbanization. This is not beneficial for the implementation of sustainable development strategies in developing countries. In order to curb and reverse the deterioration trend of the water ecological environment from the source, in 2013, the Ministry of Water Resources successively issued several notices and opinions aimed at accelerating the construction of WEC and implemented a national WEC pilot project, putting forward the concept of WEC.

Furthermore, in view of the fact that the urbanization rate of the permanent resident population in developing countries is lower than that in developed countries, such as the United States, the United Kingdom, and Germany, and the urbanization rate of the registered population in developing countries is even lower, according to the law of the Northam curve and the urbanization development experience of developed countries, the urbanization of developing countries will maintain a rapid growth rate over the next two decades. At the same time, according to the World Water Development Report released by the United Nations, with the frequent occurrence of extreme weather events in recent years, the risk of climate change has increased sharply, and this will affect the quantity and quality of the water supply required to meet basic human needs, thus compromising the safety of drinking water for billions of people. According to the report, by 2030, the world could face 40% water shortages due to rapid urbanization, economic growth, and inequality. The problem is particularly acute in sub-Saharan Africa, Latin America, the Caribbean, Asia, the Pacific, Western Asia, and North Africa. In addition, the 2030 Agenda for Sustainable Development and the Paris Agreement on Climate Change imply that water resources play important roles in the challenges of poverty eradication and sustainable development, as well as in the challenges of climate change mitigation and adaptation. Thus, in the context of global climate change, determining how to alleviate water shortages, enhance the water resource utilization efficiency, improve the quality of water environments, and maintain the water ecological balance in a reasonable and effective way during the process of urbanization has become an urgent problem to be solved by all countries worldwide.

At present, scholars are mainly focused on the measurement of NU and the WEC level and the coordination relationship between NU and ecological civilization to alleviate energy shortages, enhance energy utilization efficiency, improve environmental quality, and maintain ecological balance. However, there have been few studies on the coordination relationship between NU and WEC and its driving factors.

This paper takes the YREB of China as a case study to discuss the coordination relationship between NU and WEC and its driving factors. By the end of 2020, the YREB accounted for about 22% of China's land area, was home to nearly 43% of the country's population, and created 46.4% of the country's GDP (data source: China Statistical Yearbook). It has an indispensable strategic position in China's regional economic development. The urbanization rate of permanent residents in the YREB increased from 51.77% in 2011 to 64.06% in 2020, with an average annual growth rate of 1.23% (data source: China Statistical Yearbook), which is higher than the national average during the same period. However, against the background of the one-sided pursuit of urban expansion and the blind development of land, problems, such as water pollution, unavailability of water resources caused by pollution [3–5], and the degradation of the water ecological environment, have also emerged in the YREB, which have led to significant challenges for the urbanization development of the

YREB in the future and have seriously restricted the sustainable economic development of the YREB. Determining how to improve the level of WEC during the process of urbanization, control water pollution, and overcome the unavailability of water resources caused by pollution have also become urgent problems to be solved in the YREB. At the 18th National Congress of the Communist Party of China in 2012, the concept of NU was clearly put forward. NU describes the four-in-one urbanization of the economy, population, land, and society. It is intended to overcome the shortcomings of traditional urbanization methods that emphasize speed over quality, pay more attention to the connotation construction of urbanization, strive to improve the quality of urbanization, and coordinate harmonious development between man and nature. The integrated development between urbanization and WEC is an important part of promoting harmonious development between man and nature and has become an important topic associated with China's economic development and ecological civilization construction. Therefore, this paper analyzes the coordination degree of NU and WEC in the YREB, which can help to control water pollution, solve the problem of water resource unavailability caused by pollution, and improve the level of WEC in the YREB. This information can also be used to test the effect of the implementation of the NU strategy from the perspective of the construction of WEC [6]. It provides a reference for other countries and regions similar to the YREB to promote the harmonious coexistence of humans and water.

The literature on NU and WEC mainly includes three aspects: First is the measurement of the NU level. Since the concept of "NU" was clearly put forward at the 18th National Congress of the CPC (the Communist Party of China), the measurement of the NU level in various regions has been studied by scholars. Xiong and Xu, Yang and Sun, Yang et al., Liang et al., and Zhao et al., respectively, used China's provinces, western regions, central regions, the YREB, and the Yellow River Basin as case studies and found that the NU level in different regions was heterogeneous, showing a trend of being high in the east and low in the west [7–11]. Second is the evaluation of WEC. Scholars have studied the measurement of WEC in various regions using different methods. Yu et al., Qi and Song, Tian et al., Zhang and Wang, and Fang et al., respectively, used the YREB, the Yellow River Basin, the Pearl River Delta Region, Jiangxi Province, and Zhongshan City as case studies, constructed an indicator system, and used the connection number method, the set pair analysis method, the AHP (Analytic Hierarchy Process), fuzzy comprehensive method, and other methods to analyze WEC [12–16]. The results show that, since the pilot construction of a WEC city, the WEC level in China has gradually improved, and the sewage treatment efficiency, water efficiency, and total control of water use have significantly improved [17–26]. Third is the analysis of the coordination of NU and ecological civilization. Zambon et al., Ariken et al., Botequilha-Leito, and Díaz-Varela, Al-Mulali et al., and Irfan and Shaw analyzed the coordination degree of NU and the ecological environment, ecological resilience, ecological degradation, environmental pollution, and ecological efficiency in different countries [27–31], and most scholars found that the coordination degree was the largest in Eastern China, followed by Western China, and the coordination degree in Central China was the lowest. However, the coordination degrees of Western China and Central China are on the rise. The coordination degree of Eastern China has declined somewhat due to a population surge, the lack of land space optimization, and the ecological environment carrying capacity, which has exceeded the threshold in recent years. In addition, the literature most closely related to the paper are studies on the impacts of urbanization on water use [32–36], water security [37–41], water quality [42–46], the water footprint [47–49], and the water ecological environment [2,50,51].

Most of these studies found that (1) urbanization is affected by its respective development stage, and there are nonlinear relationships with water use, water security, water quality, the water footprint, and the water ecological environment, (2) the impacts of urbanization on water use, water security, water quality, the water footprint, and the water ecological environment are heterogeneous across regions. However, water resources and the water ecological environment are only components of WEC. Therefore, studies on

the impacts of urbanization on water use, water security, water quality, water footprint, and the water ecological environment cannot be equivalent to a coordination analysis of urbanization and WEC.

To sum up, the existing literature mainly discusses the measurement of the NU level and the WEC level and studies the coordination of NU and ecological civilization, while very few studies have investigated the coordination relationship of NU and WEC, and few scholars have analyzed the coordination impact factors [52]. Zhu et al. only used the unit root and Granger causality test methods to study the relationship between NU and WEC and did not study coordination and its impact factors [53]. Accordingly, to enrich the existing literature and make up for the shortcomings in the literature, this paper uses the YREB of China as a case study, constructs a more comprehensive indicator system of NU and WEC, and uses the state coordination function to examine the coordination of NU and WEC from 2011 to 2020. Further, we construct a two-way fixed-effects model to empirically analyze the driving factors and the heterogeneity underlying the coordination of NU and WEC. This study helps to improve the theory of NU and enrich the research content on the theory of ecological civilization. Therefore, compared with previous studies, this paper may have marginal contributions from three aspects: The use of research methods, the selection of cases, and the analysis of heterogeneity.

## 2. Research Design and Methods

### 2.1. Research Area

In 2014, the development of the YREB became part of China's national strategy to build a golden economic belt with a more beautiful ecology, more convenient transportation, a more coordinated economy, a more unified market, and more scientific mechanisms based on the Yangtze River. The YREB is one of the most important economic regions with the largest population density in China and is very important for China's regional development. It is a region with the Yangtze River as the link, the urban economic zone as the basic unit, and the river basin as the basis. It is located at latitudes 21–35° north and longitudes 97–123° east, spanning the three major regions of Eastern, Central, and Western China, starting from Shanghai in the east and ending at Yunnan in the west. The YREB covers 11 provinces, as shown in Figure 1. It is endowed with rich water resources. The total amount of water resources is 961.6 billion $m^3$, accounting for 42% of the national water resources, 20 times that of the Yellow River Basin. At the same time, there are many rivers and lakes in the YREB, including 45 tributaries with a drainage area of more than 10,000 $km^2$, 7 first-level tributaries with a drainage area of more than 80,000 $km^2$, and important lakes, such as Dongting Lake, Poyang Lake, Taihu Lake, Chaohu Lake, and Hongze Lake. However, problems, such as the unavailability of water resources caused by pollution, insufficient per capita water resources (this means not everyone has enough water), and water eutrophication restrict the construction process of the ecological civilization demonstration area [2].

### 2.2. Construction of an Indicator System

We construct an indicator system for NU. According to the National New Urbanization Plan (2014–2020), referring to the existing research results [54–57], and based on the situation of urbanization, this paper constructs an NU evaluation system with four aspects: Economic urbanization, population urbanization, land urbanization, and social urbanization.

Firstly, economic urbanization is an important support factor for the development of NU and has significant interaction with economic growth. NU is a new driving force for economic growth, and economic growth is the foundation of NU. With the agglomeration of production factors in the process of urbanization, economies of scale will be generated, and urban residents' incomes will be improved. At the same time, according to the Peddy–Clark theorem, the process of industrial structure optimization is the transformation from agriculture to manufacturing and then from manufacturing to service, which can not only achieve positive industrial interactions but also remove the problems associated with

traditional urbanization development. In addition, greater economic openness enables urban development to share the fruits of its knowledge and allows technology spillover to a greater extent. The intensity of investment plays a positive role in promoting urbanization construction, which can improve the degree of urban accommodation and the quality of life and employment. Therefore, economic urbanization is evaluated from the perspectives of economic growth (considering that the per capita GDP excludes the influence of population size and is a relative indicator, this paper uses the per capita GDP as an indicator to measure economic growth), urban residents' incomes (this paper uses the per capita disposable income of urban residents to measure urban residents' incomes), industrial structure optimization (it is difficult to fully describe changes in the industrial structure by using the proportion of the manufacturing industry in the regional GDP or the proportion of the service industry in the regional GDP alone. This paper uses the proportion of the nonagricultural industry in the regional GDP to measure industrial structure optimization), the economic opening degree (indicators that can be used to measure the economic opening degree include the proportions of the total import and export volumes in the regional GDP and the actual utilization of foreign capital per capita. However, considering the consistency of data caliber and the data availability, this paper uses the actual utilization of foreign capital per capita to measure the economic opening degree), and the investment intensity (this paper uses the investment in fixed assets by the whole society to measure the urban investment intensity). Therefore, the following secondary indicators are used to measure economic urbanization: The per capita GDP, the per capita disposable income of urban residents, the proportion of the nonagricultural industry in the regional GDP, the actual utilization of foreign capital per capita, and the investment in fixed assets by the whole society.

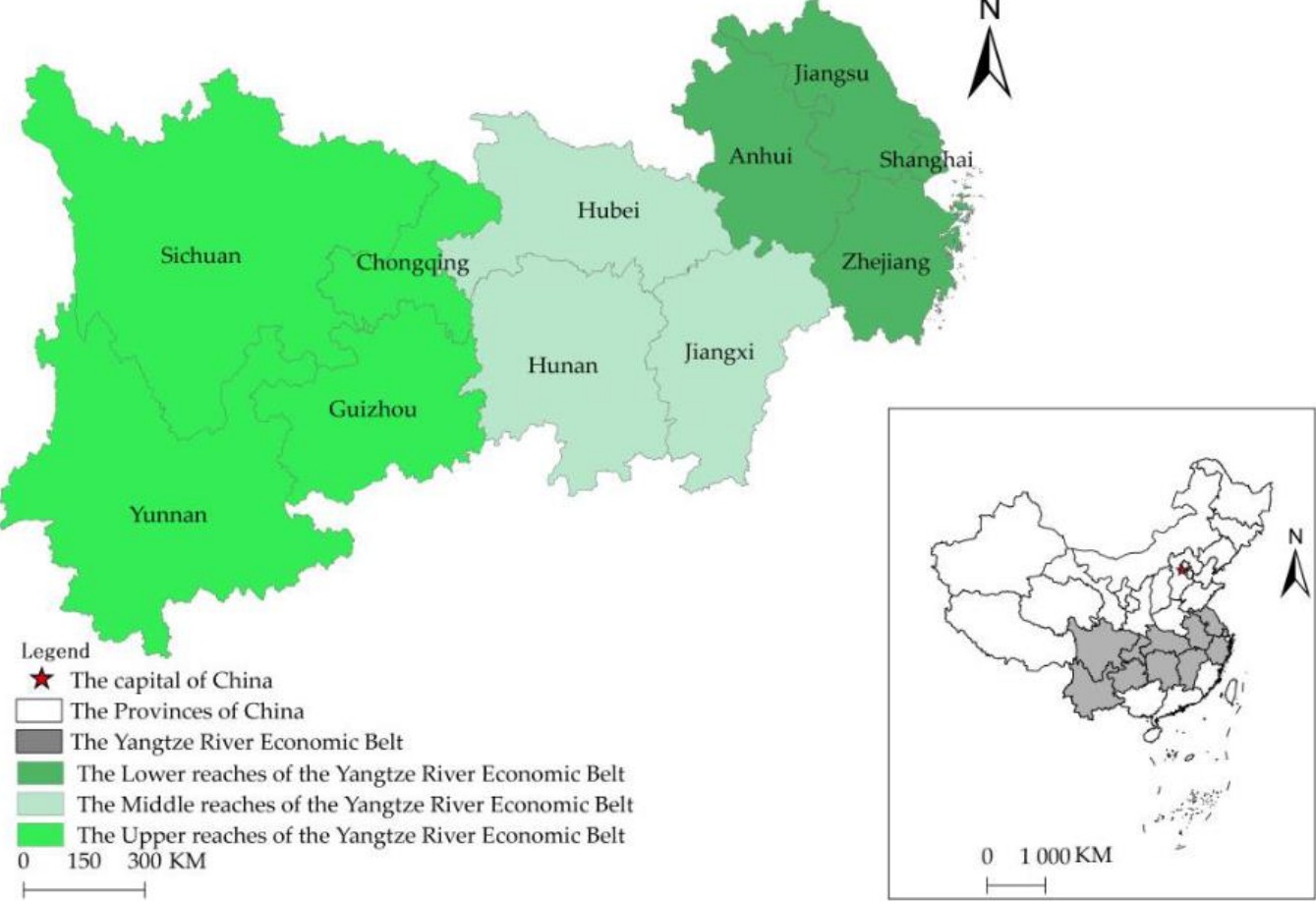

**Figure 1.** Map of The Yangtze River Economic Belt.

Secondly, population urbanization is the main driving force of NU and the core part of NU. Population agglomeration from rural to urban is the most direct sign of the urbanization process, which directly reflects the scale of urbanization development. This agglomeration effect is not only reflected in the growth of the population size but also in the growth of the urban population density, nonagricultural employment, and the urban population quality. At the same time, in order to maintain the coordinated development of the population urbanization quality and the population urbanization speed, special attention should be paid to the stability of nonagricultural employment as the basis of farmer citizenization. Therefore, population urbanization is evaluated from the perspectives of the urban population size (considering the special household registration system in China and the lack of statistics on the urbanization rate of the registered population, this paper uses the urbanization rate of the permanent population to measure the urban population size), the nonagricultural population (considering that the development of NU is based on nonagricultural employment, this paper uses the proportion of employees in the nonagricultural industry to measure the nonagricultural population), the population quality (the indicators to measure the population quality mainly include the average number of years of schooling, the average number of students in colleges and universities per $10^5$ people, and the per capita expenditure on education. Considering that the average number of years of schooling is calculated based on a population sampling survey, the results may be biased, and the data cannot be obtained at the prefecture city level. In this paper, the average number of students in colleges and universities per $10^5$ people and the per capita expenditure on education are used to measure the population quality), and employment stability (at present, the mainstream method is to use the urban registered unemployment rate to indirectly measure employment stability, which is also the case in this paper). Therefore, the urbanization rate of the permanent population, the proportion of employees in the nonagricultural industry, the average number of students in colleges and universities per $10^5$ people, the per capita expenditure on education, and urban registered unemployment rate are used to measure population urbanization.

Thirdly, land urbanization is the main carrier of new-type urbanization. It emphasizes deep exploitation of the potential of existing construction land under the concept of sustainable development. On the one hand, it promotes the reuse of urban stock space, on the other hand, it promotes the intensive development of urban construction land and improves the ecological benefits of land to provide the guarantee necessary for the achievement of new livable ecological urbanization. Therefore, land urbanization was evaluated by the land use intensity (the indicators to measure land use intensity include the population density, proportion of the built-up area in the urban area, plot ratio, and building density. Considering that the latter two are limited to the land use intensity of building land and do not include land used for regional transportation facilities, public facilities, and agricultural and forestry land, they cannot fully represent the expansion of the land scale against the background of NU. In this paper, the urban population density and the proportion of the built-up area in the urban area were used to measure the land use intensity), the land ecological benefit (the indicators used to measure the land ecological benefit include the per capita green park area, the green coverage rate of built-up areas, and the forest coverage rate. Considering that the per capita green park area is a relative indicator and excluding the influence of population size, this paper uses the per capita green park area to measure the land ecological benefit), and the land use efficiency (in this paper, the per capita road area and per capita built-up area are used to measure the land use efficiency). Therefore, the paper uses the following five indicators to measure land urbanization: The urban population density, the proportion of the built-up area in the urban area, the per capita green park area, the per capita road area, and the per capita built-up area.

Finally, social urbanization is an important embodiment of NU that puts people first. It can accelerate the implementation of equal access to basic public services, promote balanced urban and rural development, and create a harmonious social environment. Social urban-

ization can promote the process of citizenization of the migrant agricultural population, gradually allow the migrant agricultural population to enjoy the same treatment as urban population in terms of medical treatment, culture, transportation, and infrastructure, and allow them to fully integrate into urban society. Therefore, social urbanization is evaluated by the availability of basic public services, such as medical treatment, culture, transportation, and the gas penetration rate. There are many indicators that can be used to measure basic public services. However, considering the availability of data and the usefulness and universality of indicators, the paper uses the following indicators to measure social urbanization: The number of health technicians per $10^3$ people, the number of beds in medical and health institutions per $10^4$ people, the number of books collected by public libraries per capita, the number of bus and electric vehicles per $10^4$ people, and the gas popularity rate.

On the other hand, we construct a WEC indicator system. This paper constructs the WEC evaluation system using three aspects: Pressure, status, and response.

Firstly, pressure refers to the destruction of the water ecological environment caused by human social activities and the industrialization process, and more attention is paid to the economic efficiency of water resources in the face of a strict water resource management system. The former is measured by indicators such as the chemical oxygen demand discharge per $10^4$ \$ of the GDP, ammonia nitrogen emissions per $10^4$ \$ of the GDP, the blue water quality index, the fertilizer application intensity, and the groundwater exploitation coefficient, while the latter is measured by indicators such as the water consumption per $10^4$ \$ of the GDP and the water consumption per $10^4$ \$ of added value due to manufacturing.

Secondly, status refers to the situation of natural resources and environmental quality under pressure. In terms of water resources, it refers to the natural endowment of water resources, the water supply capacity, and the water quality [58]. This paper uses the following indicators to measure status: The proportion of surface water in water resources, the proportion of groundwater in water resources, the development and utilization rate of water resources, the urban water popularity rate, and the per capita water resources.

Finally, the response usually refers to the contribution made by humans to the reduction of pressure on the water ecological environment, aiming to improve the urban sewage treatment capacity, the flood control and drainage capacity, the soil conservation and water storage capacity, and water-saving awareness. The paper uses the following indicators to measure this factor: The urban sewage treatment rate, the density of drainage pipes in built-up areas, the control rates of water and soil losses, the proportion of investment into environmental pollution treatment in the GDP, the green coverage rate of built-up areas, the percentage of forest cover, the proportion of the total wetland area in the land area, and the proportion of the water-saving irrigation area (see Table 1 for the specific indicator system). The principal component analysis method was used to process the data, and the Z-score method was used to analyze the reverse indicator.

**Table 1.** Indicator system.

| First-Level Indicators | Second-Level Indicators | Third-Level Indicators | Unit of Measurement | Effect Direction |
|---|---|---|---|---|
| new urbanization | Economic urbanization | Per capita GDP | $ | + |
| | | Per capita disposable income of urban residents | $ | + |
| | | Proportion of the nonagricultural industry in the regional GDP | % | + |
| | | Actual utilization of foreign capital per capita | $ | + |
| | | Investment in fixed assets by the whole society | $10^8$ $ | + |
| | Population urbanization | Urbanization rate of the permanent population | % | + |
| | | Proportion of employees in the nonagricultural industry | % | + |
| | | Average number of students in colleges and universities per $10^5$ people | person | + |
| | | Per capita expenditure on education | $ | + |
| | | Urban registered unemployment rate | % | − |
| | Land urbanization | Urban population density | person/km$^2$ | + |
| | | The proportion of the built-up area in the urban area | % | + |
| | | Per capita green park area | m$^2$ | + |
| | | Per capita road area | m$^2$ | + |
| | | Per capita built-up area | m$^2$ | + |
| | Social urbanization | Number of health technicians per $10^3$ people | person | + |
| | | Number of beds in medical and health institutions per $10^4$ people | pcs | + |
| | | Number of books collected by public libraries per capita | copy | + |
| | | Number of bus and electric vehicles per $10^4$ people | pcs | + |
| | | Gas popularity rate | % | + |
| Water ecological civilization | Pressure | Chemical oxygen demand discharge per $10^4$ $ of GDP | kg | − |
| | | Ammonia nitrogen emission per $10^4$ $ of GDP | kg | − |
| | | Blue water quality index | − | − |
| | | Fertilizer application intensity | kg/hm$^2$ | − |
| | | Groundwater exploitation coefficient | − | − |
| | | Water consumption per $10^4$ $ of the GDP | m$^3$ | − |
| | | Water consumption per $10^4$ $ of added value due to manufacturing | m$^3$ | − |
| | Status | Proportion of surface water in water resources | % | + |
| | | Proportion of groundwater in water resources | % | + |
| | | Development and utilization rate of water resources | % | + |
| | | Urban water popularity rate | % | + |
| | | Per capita water resources | m$^3$ | + |
| | Response | Urban sewage treatment rate | % | + |
| | | Density of drainage pipes in built-up areas | km/km$^2$ | + |
| | | Control rates of water and soil losses | % | + |
| | | Proportion of investment into environmental pollution treatment in the GDP | % | + |
| | | The green coverage rate of built-up areas | % | + |
| | | Percentage of forest cover | % | + |
| | | Proportion of the total wetland area in the land area | % | + |
| | | Proportion of the water-saving irrigation area | % | + |

Note: "+" indicates that positive correlation, "−" indicates that negative correlation.



*2.3. Research Methodology*

2.3.1. State Coordination Function

Most of the existing literature is based on the coupling degree model or the coupling coordination degree model, which gives the same weight to the urbanization system and the ecological civilization system when analyzing the coupling coordination of the two. However, this may ignore the correlation between the weights of the two systems over time, resulting in errors in the results [52]. The state coordination function is established by the concept of the membership degree in fuzzy mathematics, which overcomes the deviation caused by artificial weights in the traditional coupling coordination degree model [59]. Considering this, this paper uses the state coordination function to analyze the degrees of static and dynamic coordination between NU and WEC. The calculation formula is as follows:

$$U(i/j) = \exp\left[-\frac{(x_i - x_i')^2}{s^2}\right] \tag{1}$$

where $U(i/j)$ is the state coordination coefficient of the NU system to the WEC system. $x_i$ is the actual value of NU, $x_i'$ is the coordination value of NU required by the WEC system, and $s^2$ is the actual variance. The closer the actual value is to the coordination value, the higher the $U(i/j)$, and the higher the coordination degree of NU to WEC. However, this equation cannot measure the mutual coordination degree $U(i,j)$. The mutual coordination degree can be obtained by the following formula:

$$U(i,j) = \frac{\min\{U(i/j), U(j/i)\}}{\max\{U(i/j), U(j/i)\}} \tag{2}$$

where the higher the $U(i,j)$, the higher the mutual coordination degree of NU and WEC in the YREB, and the lower the $U(i,j)$, the lower the mutual coordination degree. However, $U(i,j)$ is only the degree of static coordination at a certain point in time. Generally, the value of $U(i,j)$ ranges from 0 to 1, where $0 \leq U(i,j) \leq 0.3$ indicates that the coordination degree is low and the two are not coordinated, $0.3 < U(i,j) \leq 0.5$ indicates that they are basically uncoordinated, $0.5 < U(i,j) \leq 0.8$ indicates they are basically coordinated, and $0.8 < U(i,j) \leq 1$ indicates that the coordination degree is high, and the two develop in a coordinated way (see Table 2).

**Table 2.** Classification of degree of static coordination.

| Serial Number | Static Coordination Degree | Coordination Type |
|---|---|---|
| 1 | $0 \leq U(i,j) \leq 0.3$ | Incoordination |
| 2 | $0.3 < U(i,j) \leq 0.5$ | Basic incoordination |
| 3 | $0.5 < U(i,j) \leq 0.8$ | Basic coordination |
| 4 | $0.8 < U(i,j) \leq 1$ | Coordination |

Since NU and WEC are dynamic processes in time series, the degree of dynamic coordination $U(i,j/t)$ of the two needs to be calculated. The calculation formula is as follows:

$$U(i,j/t) = \frac{1}{T}\sum_{i=0}^{T-1} U(i,j/(t-i)) \tag{3}$$

where the higher the $U(i, j/t)$, the higher the degree of dynamic coordination between NU and WEC in the YREB. Generally, the value of $U(i, j/t)$ ranges from 0 to 1. If, at two different times, $t_1$ and $t_2$, $t_1 < t_2$, there is $U(i, j/t_1) < U(i, j/t_2)$, which indicates that NU and WEC have been on a coordinated development path.

2.3.2. Model Construction

The coordination degree of NU and WEC is affected by many factors. By referring to previous relevant research [11,60,61], we conducted an empirical study by setting the degree of static coordination (*Sc*) as the explained variable and using the following explanatory variables: The opening-up degree (*Op*), the upgrade of the industrial structure (*In*), the marketization level (*Ma*), human capital (*Ed*), foreign direct investment (*Fd*), population growth (*Po*), and the urban–rural income gap (*Ur*). The logarithmic forms of all variables were included in the model to effectively solve the problem of the estimation bias caused by heteroscedasticity and multicollinearity. The model is as follows:

$$\ln Sc_{it} = C + \beta_1 \ln Op_{it} + \beta_2 \ln In_{it} + \beta_3 \ln Ma_{it} + \beta_4 \ln Ed_{it} + \beta_5 \ln Fd_{it} + \beta_6 \ln Po_{it} + \beta_7 \ln Ur_{it} + \mu_i + \phi_t + \varepsilon_{it} \quad (4)$$

where $i$ is the province (city) of the YREB, and $t$ is the year. There are 11 sample provinces and cities, and the sample period is 2011–2020.

2.3.3. Variable Measurement and Data

First, the total foreign trade/GDP, the added value of the tertiary industry/GDP, the marketization index, the local financial education expenditure/local general budget expenditure, the achieved foreign direct investment/GDP, the natural growth rate of the population, and the Theil index were used to measure the opening-up degree, the upgrade of the industrial structure, the marketization level, human capital, foreign direct investment, population growth, and the urban–rural income gap, respectively [62–66]. The original data on the blue water quality index were obtained from the "blue map" issued by the Institute of Public and Environmental Affairs. The original data for the other indicators and variables were obtained from the China Statistical Yearbook and the provincial statistical yearbooks, the China Population and Employment Statistical Yearbook, the China Environmental Statistical Yearbook, the China Water Resources Bulletin, and the provincial water resources bulletins. The study used linear interpolation and mean methods to fill in missing data. Table 3 shows the descriptive statistical results of the variables.

**Table 3.** Descriptive statistical results of variables.

| Variable | Mean | Standard Deviation | Minimum | Maximum |
|---|---|---|---|---|
| ln*Sc* | 0.578 | 0.127 | 0.124 | 0.693 |
| ln*Op* | −1.692 | 0.938 | −3.611 | 0.387 |
| ln*In* | 3.878 | 0.149 | 3.523 | 4.292 |
| ln*Ma* | 2.132 | 0.219 | 1.420 | 2.479 |
| ln*Ed* | 2.806 | 0.129 | 2.397 | 3.049 |
| ln*Fd* | 0.548 | 0.843 | −1.775 | 1.504 |
| ln*Po* | 1.607 | 0.621 | −2.120 | 2.216 |
| ln*Ur* | 0.090 | 0.043 | 0.019 | 0.205 |

## 3. Research Results

### 3.1. Evaluation Results

First, as shown in Table 4, the growth rate of the NU level has always been positive and shows a trend of fluctuating upward in the YREB. The reason for this may be that early urbanization construction was in the exploratory stage in which attention is paid to the speed of urbanization, blindly expanding the scale of cities, ignoring the connotation of urbanization construction, and failing to improve the urbanization quality while undergoing extensive urbanization. Since the 18th National Congress of the Communist Party of China (CPC), the country has attached great importance to the construction of NU and has put forward many related policies. However, due to the lag in the implementation and effectiveness of the policies, the development of NU is still insufficient. Table 4 shows that the growth rate of NU in the YREB and its regions (the upper reaches, the middle reaches, and the lower reaches) reached its maximum value in 2019. The reason for this

may be that since the implementation of the National New Urbanization Plan in 2014, the marginal effects of various policies on NU construction have gradually increased, and more attention has been paid to the connotation construction of urbanization, thus gradually improving the urban infrastructure and public service system. The utilization rates of urban space and the ecological environment have also been improved. From a spatial point of view, the average annual growth rates of NU in the upper, middle, and lower reaches are 0.344, 0.359, and 0.283, respectively, showing a trend of sequential decline in the middle, upper, and lower reaches, which may be related to policy support and the economic scale. The middle reaches are the pilot area for NU in Central and Western China and the key region for implementing the "Rise of Central China" strategy. They have shown significantly enhanced NU in recent years. In addition, the middle reaches are affected by the economic radiation of the lower reaches, and their economic scale is higher than that of the upper reaches, thus the investment in NU construction is relatively large. Additionally, the importance of the middle reaches is due to their unique geographical advantages (they are mainly plains), the relatively concentrated population, and the urbanization development potential. It is worth noting that the average annual growth rate of the lower reaches is lower than that of other regions. The reason for this is that the lower reaches, as some of the most developed economic regions in China, have accelerated NU construction by relying on the factor agglomeration effect, human capital effect, foreign investment effect, and economic scale effect, which means that the optimization spaces of infrastructure, public services, and the living environment tend to be saturated. As a result, the NU growth rate is lower than that of other regions, resulting in the problem of insufficient momentum for NU construction.

**Table 4.** The growth rate of NU and WEC.

| Year | Yangtze River Economic Belt | | The Upper Reaches | | The Middle Reaches | | The Lower Reaches | |
|---|---|---|---|---|---|---|---|---|
| | $\Delta NU$ | $\Delta WEC$ | $\Delta NU$ | $\Delta WEC$ | $\Delta NU$ | $\Delta WEC$ | $\Delta NU$ | $\Delta WEC$ |
| 2012 | 0.322 | 0.105 | 0.381 | 0.131 | 0.383 | −0.002 | 0.218 | 0.161 |
| 2013 | 0.350 | 0.143 | 0.406 | 0.127 | 0.263 | 0.153 | 0.360 | 0.153 |
| 2014 | 0.317 | 0.127 | 0.395 | 0.111 | 0.366 | 0.105 | 0.203 | 0.159 |
| 2015 | 0.285 | 0.180 | 0.281 | 0.138 | 0.316 | 0.110 | 0.266 | 0.275 |
| 2016 | 0.327 | 0.213 | 0.377 | 0.186 | 0.322 | 0.105 | 0.282 | 0.321 |
| 2017 | 0.361 | 0.193 | 0.374 | 0.121 | 0.333 | 0.198 | 0.369 | 0.263 |
| 2018 | 0.355 | 0.147 | 0.383 | 0.104 | 0.378 | 0.195 | 0.309 | 0.153 |
| 2019 | 0.473 | 0.221 | 0.455 | 0.252 | 0.582 | 0.036 | 0.410 | 0.327 |
| 2020 | 0.143 | 0.281 | 0.042 | 0.162 | 0.293 | 0.096 | 0.131 | 0.539 |
| Mean | 0.326 | 0.179 | 0.344 | 0.148 | 0.359 | 0.111 | 0.283 | 0.261 |

Secondly, from Table 4, it can be found that the growth rate of the WEC level also shows a fluctuating trend and the growth rate of the WEC level in 2014 was minimal after the implementation of the pilot policy of WEC city construction. The reason for this may be that all regions were still in the exploratory stage of WEC city construction in 2014 and lacked a guiding ideology of construction and sufficient publicity of this ideology. All regions still considered economic benefits first and failed to guide citizens to develop a good awareness of environmental protection. In 2015 and 2016, the growth rate of the WEC level gradually increased, which may have been mainly due to the fact that regions gradually attached importance to construction, began to implement the strictest water resource management system, standardized the use and development of water resources, publicized the idea of WEC with the help of new media, built a water-saving society, and cultivated citizens' awareness of water conservation. In addition, in view of previous prominent problems associated with the water environmental quality, the implementation of the "4 + 1" project in 2015 and the formation of a sound sewage charging system in 2016 significantly improved the sewage treatment capacity and sewage treatment rate of all regions in the YREB in 2015 and 2016 [67]. With the reduction of the marginal effect of

the above-mentioned policies and a series of documents related to the pilot construction of WEC, the WEC growth rate fell back in 2017 and 2018 but then gradually increased to the maximum level due to the implementation of environmental protection policies and regulations, such as the Environmental Protection Tax Law, the Reform Program of the Ecological and Environmental Damage Compensation System, and the newly revised Water Pollution Prevention and Control Law. From the perspective of space, the average annual growth rate of WEC in the upper, middle, and lower reaches of the YREB is 0.148, 0.111, and 0.261, respectively, presenting the completely opposite situation to that of NU, with the trend of decreasing successively in the lower reaches, upper reaches, and middle reaches, which is consistent with the research conducted by Yu et al. and Su et al. [12,68]. This is mainly because the lower reaches of the YREB, which have a strong economic foundation and strong investment in infrastructure, have laid a good foundation for environmental protection and sewage treatment. In addition, the lower reaches of the YREB rely on the population siphon effect to gather high-end talents, accelerating the transformation of scientific research achievements, which aids in the construction of WEC. Moreover, in order to comply with the requirements of the national development strategy, the lower reaches of the YREB have actively introduced high-tech and emerging industries while transferring their chemical enterprises to the middle reaches of the YREB, continuously optimizing the industrial layout and reducing the emissions of three wastes (waste gas, wastewater, and industrial residue). As a link between Eastern and Western China, the middle reaches of the YREB, which are abundant in land resources and the labor force, bear the chemical enterprises and labor-intensive enterprises transferred from the lower reaches of the YREB, and most of these enterprises produce high-water-consumption and pollution-type products. This region has insufficient endogenous power to alleviate the water resource supply and demand problem, improve the water environment quality, and treat wastewater, which restricts the development of WEC.

### 3.2. Estimation Results of the Coordination Degree

The static coordination degree and dynamic coordination degree of NU and WEC in the YREB were explored by using the formula of the state coordination function. Table 5, Figures 2 and 3 display the results of the calculations.

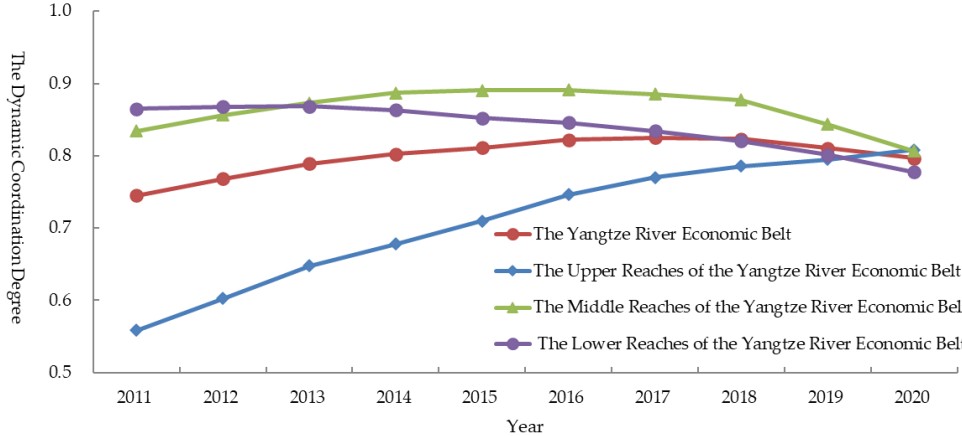

**Figure 2.** The Dynamic coordination degree.

**Table 5.** Static coordination degree.

| Year | Yangtze River Economic Belt | | The Upper Reaches | | The Middle Reaches | | The Lower Reaches | |
|------|-------|-------------------|-------|-------------------|-------|-------------------|-------|-------------------|
| | *U(i,j)* | Coordination Type | *U(i,j)* | Coordination Type | *U(i,j)* | Coordination Type | *U(i,j)* | Coordination Type |
| 2011 | 0.745 | Basic coordination | 0.559 | Basic coordination | 0.834 | Coordination | 0.865 | Coordination |
| 2012 | 0.791 | Basic coordination | 0.646 | Basic coordination | 0.879 | Coordination | 0.870 | Coordination |
| 2013 | 0.832 | Coordination | 0.738 | Basic coordination | 0.906 | Coordination | 0.870 | Coordination |

Table 5. *Cont.*

| Year | Yangtze River Economic Belt | | The Upper Reaches | | The Middle Reaches | | The Lower Reaches | |
|---|---|---|---|---|---|---|---|---|
| | *U(i,j)* | Coordination Type | *U(i,j)* | Coordination Type | *U(i,j)* | Coordination Type | *U(i,j)* | Coordination Type |
| 2014 | 0.841 | Coordination | 0.769 | Basic coordination | 0.928 | Coordination | 0.848 | Coordination |
| 2015 | 0.846 | Coordination | 0.838 | Coordination | 0.906 | Coordination | 0.808 | Coordination |
| 2016 | 0.877 | Coordination | 0.927 | Coordination | 0.894 | Coordination | 0.815 | Coordination |
| 2017 | 0.842 | Coordination | 0.913 | Coordination | 0.851 | Coordination | 0.764 | Basic coordination |
| 2018 | 0.814 | Coordination | 0.897 | Coordination | 0.820 | Coordination | 0.725 | Basic coordination |
| 2019 | 0.708 | Basic coordination | 0.870 | Coordination | 0.572 | Basic coordination | 0.648 | Basic coordination |
| 2020 | 0.671 | Basic coordination | 0.923 | Coordination | 0.476 | Basic incoordination | 0.564 | Basic coordination |

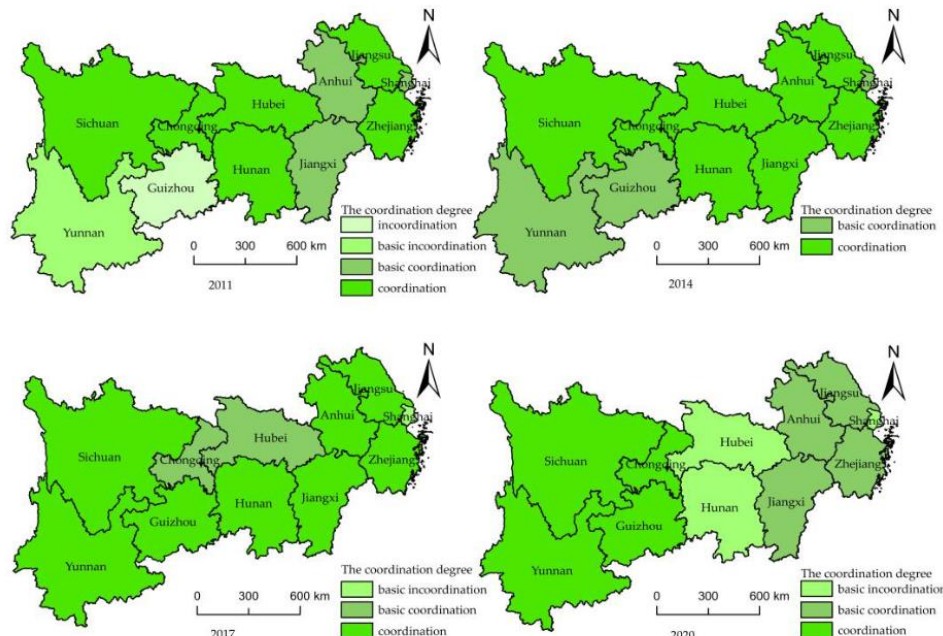

**Figure 3.** Spatial evolution pattern of the coordination degree.

First, from Table 5, it can be found that the static coordination degree was always within the range of 0.6–0.9, which indicates the basic coordination stage and the coordination stage. The static coordination degree shows a fluctuating trend of first increasing and then decreasing, which mainly indicates an evolution process of basic coordination (2011–2012) to coordination (2013–2018) and then back to basic coordination (2019–2020). This may be because, although the urbanization process was still dominated by extensive and extended development in 2011 and 2012, it led to negative effects, such as increases in water resource exploitation, the utilization rate, the groundwater extraction coefficient, and sewage and wastewater discharge and a decrease in the water utilization efficiency, the values of which are within the carrying capacity of the ecological environment, and the pressure on water resources could be relieved by the ecological self-rehabilitation function in a short time period. From 2013 to 2018, the static coordination degree was higher than 0.8, crossing the threshold and moving from the basic coordination stage to the coordination stage. This is mainly because the NU moved from an exploratory period to a development period in which the NU quality improved, the policy effect of WEC city construction gradually emerged, and the WEC level continuously increased, causing the two factors to enter the coordination stage. At the same time, the implementation of policies related to the strengthening of the construction of beautiful China, building ecological civilization demonstration zones, and support for the construction of NU in recent years have helped regions to focus on cultivating public awareness of water conservation, regulating the use of water resources, protecting water ecosystems, improving the quality of the water environment, and spreading the idea of WEC while improving the urbanization quality. However, from 2019 to 2020, NU moved back to the basic coordination stage. This may be because the NU reached a relatively high level at this time, showing a diminishing marginal effect, which had a negative impact on the

rise in WEC. In terms of the dynamic coordination degree, Figure 2 shows that the dynamic coordination degree of NU and WEC has not always been on the trajectory of coordinated development. The dynamic coordination degree from 2011 to 2017 showed a steady upward trajectory, that is, $U(i,j/t_1) < U(i,j/t_2)$, indicating that the two factors were in coordinated development during this period. Subsequently, from 2018 to 2020, the dynamic coordination degree of both decreased slightly, that is, $U(i,j/t_2) < U(i,j/t_1)$, indicating that they deviated from the trajectory of coordinated development during this period.

Secondly, from Table 5, it can be found that the static coordination degree in the upper reaches shows an upward trend, with the most obvious increase occurring through the evolution of the basic coordination stage (2011–2014) and the coordination stage (2015–2020). Although the basic coordination stage was present from 2011 to 2014, the static coordination degree increased from 0.559 in 2011 to 0.769 in 2014, nearly exceeding the threshold value required by the coordination stage. The reason for this is that the upper reaches have abundant precipitation and sufficient per capita water resources. At the same time, mountains and hills dominate the region, and there is a low industrialization level and slow urbanization process, which have not had a significant impact on the ecological environment. This area was in the coordination stage from 2015 to 2020, and its static coordination degree also increased from 0.838 in 2015 to 0.923 in 2020, which may be attributed to the release and implementation of the National New Urbanization Plan (2014–2020), China has vigorously supported the construction of NU in the upper reaches, which has greatly improved the infrastructure and public service supply, continuously optimized the industrial structure, led to a focus on the development of the tertiary industry, and changed the traditional economic development model in this region. Meanwhile, due to the low population density in the upper reaches, the few heavy-chemical and polluting enterprises, and the rich endowment of water resources, the development, and use of water resources are always maintained at a reasonable level, which significantly improves the coordination degree between the two. Both the middle and lower reaches were in the coordination stage from 2011 to 2016, and then the coordination degree of the lower reaches took the lead by changing in 2017. This may be because the lower reaches ignored the rational allocation of various factors in the process of accelerating the transformation of the industrial structure, leading to the obvious negative externality effect of urbanization construction. However, with the long-term mechanism of WEC established in the lower reaches in the early stage, the lower reaches were in the basic coordination stage. Subsequently, the static coordination degree of the middle reaches also decreased from coordination to basic coordination or even basic incoordination in 2019. This may be because the middle reaches have been a cluster of industrial transfer in recent years, and most of the transferred enterprises have been labor-intensive enterprises, heavy chemical enterprises, and low-quality foreign-funded enterprises, resulting in a surge in industrial and residential water consumption and an increase in the discharge of pollutants in wastewater. In order to solve these problems, it is necessary to accelerate the urbanization process and improve the construction of infrastructure, especially by increasing drainage pipes, sewage treatment facilities, unconventional water collectors, water-saving appliances, and other facilities to improve the sewage treatment rate and water resource utilization efficiency, but these measures to improve the WEC level are not able to achieve the expected results in a short time period, and the improvement of this infrastructure and the promotion of subsequent projects will consume a significant amount of water, which is why the WEC growth rate in the middle reaches is the lowest, while the growth rate of the NU level is the largest. In terms of the dynamic coordination degree, Figure 2 shows that the dynamic coordination degree of the upper reaches has a linear upward trend, indicating that NU and WEC in the region have been on a coordinated development trajectory. The dynamic coordination degree in the middle reaches was on a coordinated development trajectory from 2011 to 2016 and then deviated from the coordinated development trajectory from 2017 to 2020. The dynamic coordination degree in the lower reaches had a coordinated development trajectory from 2011 to 2013 and deviated from the coordinated development trajectory for the rest of the period.

Third, from Figure 3, it can be found that the static coordination degree in 2011 included four stages. Guizhou and Yunnan provinces were in the incoordination and basic incoordination stages, accounting for 18.18% of the study sample, which may be due to the fact that the NU levels of both were lower compared to the WEC levels of Guizhou and Yunnan. Provinces and cities in the basic coordination stage included Shanghai, Anhui, and Jiangxi, accounting for 27.27% of the study sample, while provinces and cities in the coordination stage were Zhejiang, Jiangsu, Hubei, Hunan, Chongqing, and Sichuan, accounting for 54.55% of the study sample. It is worth noting that Shanghai is not in the stage of coordination. The reason for this may be that, as the leader of the YREB, Shanghai has a higher NU level than that of other provinces and cities, depending on its geographical advantages and relatively complete infrastructure and public service system. However, in the process of rapid urbanization, economic growth, population concentration, and social development have led to increases in residential water, industrial production water, and ecological water replenishment, resulting in a higher pressure on the water ecological environment than that of provinces and cities in the coordination stage. In 2014 and 2017, the static coordination degree was further improved, and there were no uncoordinated provinces and cities, most of which were in the coordination stage, accounting for 72.72% of the study sample. This shows that the provinces and cities in the YREB formed the agglomeration effect of coordinated development by relying on the dividends of the National New Urbanization Plan and the pilot policy of WEC city construction during this period. By 2020, the difference in the static coordination degree among the provinces and cities gradually became more pronounced. Chongqing, Guizhou, Sichuan, and Yunnan were in the coordination stage, mainly because of the high-water environment quality in these provinces and cities and the continuous improvement of the government's capacity for water ecological protection and water environmental governance. In addition, the population in the upper reaches has been continuously concentrated in the Yangtze River Delta in recent years, which has greatly eased the pressure on water use. However, Jiangxi, Hunan, and Hubei have moved into stages of basic coordination or basic incoordination. The reason for this is that the middle reaches have received the transfer of a lot of heavily polluting enterprises from the coastal areas, but they do not have the geographical advantages, resource endowment, innovation level, human capital, and infrastructure of the coastal areas. In addition, these provinces are the centers of energy and manufacturing enterprises, and the added value of the manufacturing industry is obviously higher than that of other regions, which has a negative impact on the construction of WEC in the NU process.

Fourth, from Figure 3, it can be found that the spatial distribution location of the static coordination degree in the YREB in 2011 was significantly different, showing a decreasing trend from the center to the two sides. The coordination degree was higher in the middle reaches, while the coordination degree of the lower reaches was slightly lower than that of the middle reaches because Shanghai and Anhui were in the basic coordination stage, thus pulling down the coordination degree of the lower reaches. The static coordination degree of the lower reaches was the lowest because Guizhou and Yunnan were in the incoordination and basic incoordination stages, respectively, which greatly reduced the static coordination degree. In 2014, the spatial distribution difference of the static coordination degree in the upper, middle, and lower reaches gradually narrowed. Specifically, Anhui, in the upper reaches, and Jiangxi, in the middle reaches, increased from the basic coordination stage to the coordination stage, and Guizhou and Yunnan, in the lower reaches, increased from the incoordination stage and basic incoordination stage, respectively, to the basic coordination stage. In 2017, the static coordination degree of various regions changed again, with Shanghai, in the upper reaches, dropping from the basic coordination stage to the basic incoordination stage, Hubei, in the middle reaches, and Chongqing, in the upper reaches, dropping from the coordination stage to the basic coordination stage, and Guizhou and Yunnan, in the upper reaches, jumping from the basic coordination stage to the coordination stage simultaneously. In 2020, the spatial distribution of the static

coordination degree was very different, and the spatial agglomeration effect was obvious. The provinces in the upper reaches were all in the coordination stage, and the coordination degree decreased successively in the upper, lower, and middle reaches, which is completely opposite to the situation in 2011. Hubei and Jiangxi, in the middle reaches, and Anhui, Zhejiang, and Jiangsu, in the upper reaches, dropped by one level, while Hunan, in the middle reaches, dropped from the coordination stage to the basic incoordination stage.

In general, only Sichuan was in the coordination stage from 2011 to 2020. The provinces in the basic coordination stage were Anhui and Jiangxi in 2011. Although they rose to the coordination stage at one stage during the evolution process, they fell back to the basic coordination stage. Guizhou and Yunnan gradually upgraded from the initial incoordination or basic incoordination stage to the coordination stage, Zhejiang and Jiangsu reduced from the coordination stage in 2011 to the basic coordination stage in 2020, and Hunan, Hubei, and Shanghai changed from the coordination or basic coordination stage in 2011 to the basic incoordination stage in 2020. Chongqing declined from the coordination stage to the basic coordination stage and finally changed to the coordination stage.

### 3.3. Regression Results

#### 3.3.1. Baseline Regression Results

We use a mixed-effects model, fixed-effects model, two-way fixed-effects model, and random-effects model for the estimation (see Table 6 for the results). Firstly, the *p*-values of the F statistic and LM statistic were both 0.000, indicating that the estimation results of the fixed-effects model and random-effects model were better than those of the mixed-effects model. Secondly, the Hausman test also showed that the *p*-value was 0.000, indicating that the estimation result of the fixed-effects model was better than that of the random-effects model. Finally, according to the results of the LR test, it was found that the null hypothesis was rejected, indicating that the time effect was also significant, thus the estimation results of the two-way fixed-effects model were optimal. Based on this, we chose the results of this model for analysis.

**Table 6.** Baseline regression results.

| Variable | Mixed-Effects Model | Fixed-Effects Model | Two-Way Fixed-Effects Model | Random-Effects Model |
|---|---|---|---|---|
| ln$Op$ | −0.024 | −0.039 | −0.099 ** | −0.034 |
| | (0.017) | (0.043) | (0.039) | (0.029) |
| ln$In$ | −0.520 *** | −0.789 *** | 0.060 | −0.550 *** |
| | (0.101) | (0.162) | (0.286) | (0.134) |
| ln$Ma$ | 0.367 *** | 0.225 | 0.527 *** | 0.237 |
| | (0.145) | (0.177) | (0.162) | (0.240) |
| ln$Ed$ | 0.089 | 0.380 *** | 0.166 | 0.212 |
| | (0.089) | (0.127) | (0.164) | (0.138) |
| ln$Fd$ | −0.071 *** | −0.023 | −0.055 ** | −0.073 *** |
| | (0.016) | (0.039) | (0.023) | (0.023) |
| ln$Po$ | 0.046 * | 0.043 * | −0.034 * | 0.052 ** |
| | (0.025) | (0.023) | (0.018) | (0.027) |
| ln$Ur$ | −0.644 | −4.214 ** | −7.201 *** | −1.856 |
| | (0.687) | (1.497) | (1.277) | (1.427) |
| C | 1.544 ** | 2.350 ** | −0.442 | 1.680 ** |
| | (0.629) | (0.906) | (1.257) | (0.812) |
| Adjust-R$^2$ | 0.470 | 0.559 | 0.757 | — |
| Individual fixed effect | No | Yes | Yes | — |
| Time fixed effect | No | No | Yes | — |

Note: *, **, and *** indicate that the variable is significant at the level of 10%, 5%, and 1%, respectively.

First, from Table 6, it can be found that the regression coefficients of the opening-up degree and foreign direct investment were significantly negative at the 5% level, indicating that the opening-up degree and foreign direct investment are not advantageous to the

improvement of the coordination degree. The former may be due to the fact that the YREB, as the main region engaged in processing trade, has not completely changed its trade mode during the process of urbanization and is still dominated by low-value-added, labor-intensive products with high water consumption and heavily polluting chemical products, leading to increases in the total water consumption and severe water pollution. In addition, when the regions of YREB learn from the advanced urbanization construction experience of developed countries, they do not consider the influences of their own economic development level, geographical location, and historical factors, blindly pursue urbanization at a large scale and high speed and ignore the urbanization quality. The reason for the latter may be that multinational companies have transferred some polluting industries to the YREB in order to circumvent the environmental regulations of their home countries, resulting in a significant increase in undesirable outputs, such as sewage discharge, which has a negative impact on the construction of WEC. Additionally, most industries with foreign direct investment are labor-intensive industries that provide jobs for the surplus rural labor transferred to cities and promote the development of NU but also lead to increases in the water resource demand and ecological environment pressure, resulting in the unavailability of water resources due to pollution, seasonal water shortages, and a water ecosystem imbalance. The regression results are similar to those obtained by Li and Zhang [60] but contrary to those obtained by Kan et al. for Jiangxi Province in China [2].

Second, from Table 6, it can be found that the regression coefficients of the marketization level, upgrade of the industrial structure, and human capital are positive, and the latter two do not pass the significance test, indicating that the positive impacts of the two factors on the coordination degree are not significant. The reason for this may be that, although most cities have optimized their industrial structures, eliminated high-polluting industries with backward production capacities, introduced high-tech and environment-friendly enterprises, and reduced water pollution emissions and water consumption, the region is still in the industrialization stage, which is not advantageous for improving the coordination degree of NU and WEC. The latter occurs because, although the human capital level increased during the sample period, the overall level was still not high, which restricted innovation and diffusion in water saving, water use, and water pollution treatment technologies and affected the coordination degree. The regression results are similar to those obtained by Zhao et al. [11], and they are also similar to those obtained by Kan et al. for Jiangxi Province in China [2].

Thirdly, from Table 6, it can be found that the regression coefficients for population growth and the urban–rural income gap are significantly negative at the 10% level and 1% level, respectively, indicating that population growth and urban–rural income gap have significant negative impacts on the coordination degree.

### 3.3.2. Heterogeneity Analysis

There is obvious heterogeneity in the coordination degree of NU and WEC. We conducted empirical tests on the driving factors of the coordination degree in the lower reaches, middle reaches, and upper reaches, and Table 7 shows the results.

**Table 7.** Estimation results of heterogeneity.

| Variable | The Lower Reaches | The Middle Reaches | The Upper Reaches |
|---|---|---|---|
| ln$Op$ | −0.096 *** | 0.045 | 0.052 |
| | (0.033) | (0.048) | (0.083) |
| ln$In$ | −0.727 *** | 0.010 | 1.045 |
| | (0.241) | (0.427) | (0.489) |
| ln$Ma$ | −0.217 ** | 0.487 | 0.754 *** |
| | (0.110) | (0.349) | (0.113) |
| ln$Ed$ | 0.448 *** | 0.080 | 0.484 * |
| | (0.142) | (0.194) | (0.153) |
| ln$Fd$ | 0.155 *** | −0.134 *** | 0.003 |
| | (0.038) | (0.038) | (0.052) |

**Table 7.** *Cont.*

| Variable | The Lower Reaches | The Middle Reaches | The Upper Reaches |
|---|---|---|---|
| ln$Po$ | 0.018 | 0.188 *** | −0.028 |
| | (0.016) | (0.015) | (0.026) |
| ln$Ur$ | −5.247 *** | 4.066 *** | −19.968 *** |
| | (1.359) | (0.703) | (3.258) |
| $C$ | 2.712 * | −1.170 | −2.809 |
| | (1.533) | (1.224) | (1.792) |
| Adjust-$R^2$ | — | — | 0.906 |
| Individual fixed effect | — | — | Yes |
| Time fixed effect | — | — | Yes |

Note: *, **, and *** indicate that the variable is significant at the level of 10%, 5%, and 1%, respectively.

From Table 7, it can be found that the opening-up degree has had a significant negative effect on the coordination degree of NU and WEC in the lower reaches and a positive effect on the coordination degree in the middle and upper reaches, but these results did not pass the significance test. The reason for this is that the processing trade scale of high-water consumption products in foreign trade by these two regions is much smaller than that of the lower reaches, and the amounts of water resources used and sewage generated are within the range of bearing.

From Table 7, it can be found that the upgrade of the industrial structure has had a significant negative effect on the coordination degree of NU and WEC in the lower reaches and a positive effect on the coordination degree in the middle and upper reaches, but these results did not pass the significance test. This may be because the upgrade of the industrial structure in the lower reaches is more concerned with the upgrade of the internal structure of the manufacturing industry and the internal structure of the service industry, making it relatively difficult to achieve an advanced industrial structure, but the upgrade of the industrial structure in the middle and upper reaches is more concerned with upgrading from a manufacturing to a service industry, making it relatively less difficult to rationalize the industrial structure. The marketization level has significantly reduced the coordination degree of the lower reaches, significantly promoted the coordination degree of the upper reaches, and promoted the coordination degree of the middle reaches in an insignificant manner.

From Table 7, it can be found that human capital has promoted the coordination degree of the lower and upper reaches, a result that passed the significance level test at levels of 1% and 10%, respectively. It has promoted the coordination degree of the middle reaches, but this result was not significant. The reason for this is that, compared with the upper reaches, the lower reaches have more obvious advantages in terms of human capital, and the effects of scientific and technological innovation are more significant, which is advantageous for the comprehensive improvement of the water use efficiency during NU construction through industrial upgrading, making the promotion of coordination degree is more obvious. Meanwhile, the human capital in the middle reaches and the upper reaches does not differ significantly, but the natural endowment of water resources in the upper reaches is better than that in the middle reaches, which is advantageous for the construction of WEC in the upper reaches.

From Table 7, it can be found that foreign direct investment has promoted the coordination degree of the upper reaches, but this result did not pass the significance test. It has significantly promoted the coordination degree of the lower reaches and has had a significant negative effect on the coordination degree of the middle reaches. The main reason for this is that, compared with the middle reaches, the lower reaches of the YREB have a higher degree of environmental regulation, forcing low-quality and high-pollution foreign-invested enterprises to enter the middle reaches. This is not advantageous for the construction of WEC in the middle reaches.

From Table 7, it can be found that population growth has only significantly promoted the coordination degree in the middle reaches. The urban–rural income gap has had a

significant negative effect on the coordination degree of the lower and upper reaches and has significantly promoted the coordination degree of the middle reaches.

## 4. Discussion

In the process of NU, one of the urgent problems is determining how to improve the level of WEC. In this paper, we constructed an indicator system and used a state coordination function and a two-way fixed effects model to study the coordination state and its driving factors for NU and WEC in the YREB from 2011 to 2020.

First, in terms of the measurement of NU and WEC, although the research subjects chosen for this paper are different from those used by Xiong and Xu, Yu [7,69], the conclusions reached in this paper on the level of NU in the YREB are consistent with their findings. Their studies show that the level of NU in China is increasing year by year, and the gaps in the NU level in Eastern, Central, and Western China are gradually widening. However, in terms of the spatial distribution pattern, the research conclusion of this paper is inconsistent with that of Liang et al. [10], who found that the YREB decreases successively from east to west. The reason for the inconsistent conclusion may be that the selected indicators and the dimension reduction methods used were different. In addition, the measurement results for WEC in this paper are consistent with those used in conducted by Yu et al. and Su et al. [69], which indicates that the application of the PSR model in this paper to construct the WEC index is reasonable and has certain reference value.

Secondly, from the perspective of the coordination degree, the results for the static coordination degree of the YREB in this paper are consistent with the research conclusions of Han et al. and Lv et al. [70,71]. The reason for this may be that, on the one hand, with the YREB beginning to carry out NU construction, good progress has been made in all aspects of the ecological environment, which is conducive to the promotion of the sustainable development strategy. On the other hand, both the improvement of the water ecological environment and the reduction of water pollution are components of WEC, and since NU is becoming more coordinated with the improvement of the water ecological environment and the reduction of water pollution, it is obvious that NU is also becoming more coordinated with WEC. However, in terms of spatial distribution, the research conclusion of this paper for the lower reaches of the YREB is completely opposite to those of Deng et al. and Li et al. [72,73]. The reason for this is that the latter used the covariance coordination model, which includes individual parameters ($K$, $\alpha$, $\beta$) determined by the researchers themselves. For example, most scholars use the same values for $\alpha$ and $\beta$ and consider the subsystem to be equally important, having a value of 1/2. Some scholars believe that even if a value of 1/3 is used, it has little impact on the result [74]. Wang verified this idea by comparing the function image and the data and found that when the subsystem value was 2, if the $K$ value was 0.5, the coordination degree was more intensive in the high-value range [75]. Therefore, even if the comprehensive index values of the two systems are significantly different, the coordination degree will be higher, and the current situation in the lower reaches of the YREB is exactly the same.

Thirdly, from the perspective of driving factors, the opening-up degree and foreign direct investment are negatively correlated with the coordination degree, indicating that the YREB is still dominated by labor-intensive industries and has not achieved the transformation to a capital-intensive and technology-intensive area. Moreover, the "pollution paradise" hypothesis is still valid in the middle reaches of the YREB but not in the lower reaches. Based on this conclusion, developing countries that still rely on labor-intensive and pollution-intensive industries to develop their economies, such as India, Mexico, and Brazil, must avoid environmental pollution by strengthening their own environmental regulations and raising the threshold of industrial transfer. The upgrade of the industrial structure has had a significant negative effect on the coordination degree of the lower reaches of the YREB, which is consistent with the research conclusions of Deng et al. and Zhou et al. [72,76], and the upgrade of the industrial structure has promoted the coordination degree of the upper reaches and the middle reaches of the YREB. On the one hand,

this shows that it is necessary to optimize and upgrade the regional industrial structure according to the functional position of the YREB to ensure sustained and stable economic development. On the other hand, although the lower reaches of the YREB are the pilot area of China's opening-up to the outside world, a large amount of foreign capital and technology is attracted to this area due to its geographical advantages, which drives the optimization of the industrial structure. However, enterprises in the lower reaches of the YREB still have problems associated with difficult and expensive financing, which is not conducive to helping enterprises to overcome the problem of insufficient funds caused by large-scale and long-cycle R&D investment during the process of breakthrough innovation and hinders green technology innovation by enterprises. As a result, enterprises are unable to improve the efficiency of water use, reduce water environment pollution, or improve the level of WEC. Human capital plays a significant role in promoting the coordination degree of the lower and upper reaches but not in the middle reaches of YREB. This conclusion is consistent with the research conclusions of Iwami [77]. Compared with Iwami's research, this paper clarifies the promoting role of human capital on water ecological civilization in the process of NU. The urban–rural income gap has had a significant negative effect on the coordination degree of the lower and upper reaches and has significantly promoted the coordination degree of the middle reaches. This conclusion can be explained by the environmental Kuznets curve proposed by Grossman and Krueger [78]. Population growth has only significantly promoted the coordination degree in the middle reaches, which is consistent with the findings of Boamah et al. and Cropper and Griffiths [79,80]. Although population growth will cause water environmental pollution in the short term, the demographic dividend effect generated by population growth can promote scientific and technological innovation and technological progress and help to comprehensively improve the efficiency of water resource utilization during NU through industrial structure upgrading and optimization while reducing undesirable outputs.

The results of this study can help to improve the water ecological environment during the process of promoting NU and achieve coordinated development between NU and WEC in the YREB. Although this study analyzed the spatio-temporal differences and driving factors associated with the coordination degree between NU and WEC in the YREB, there are still the following shortcomings. Firstly, the indicators of NU only considered the economy, population, and spatial and social urbanization, and subsequent research could include digital urbanization derived from various industries enabled by digital technology. Secondly, reliable data are the basis for measuring WEC. However, limited by the availability of some WEC indicator data, the indicator system constructed in this study cannot fully meet the needs of WEC evaluation. In addition, due to the availability of data, this paper used provincial-level data for the research, which could have reduced the scientificity and accuracy of the evaluation results. Thirdly, due to the different cases and research perspectives, the results cannot be accurately extended to other developing countries. Finally, the coupling coordination degree model was not used in this paper to analyze the coordination degree between NU and WEC in the YREB, thus this method will be used in future research, and the conclusions will be compared with the results of this paper.

## 5. Conclusions

First, the growth rate of NU and WEC in the YREB shows a fluctuating upward trend. There is significant heterogeneity between regions. Specifically, the growth rate of NU decreases from the middle reaches to the upper reaches and then to the lower reaches. The growth rate of WEC shows a distribution pattern of decreasing from the lower reaches to the upper reaches and then to the middle reaches.

Second, the static coordination degree of NU and WEC in the YREB shows a trend of fluctuating upward and then falling, and the dynamic coordination degree deviated from the coordinated development trajectory from 2018 to 2020. The static coordination degree in the upper reaches shows an upward trend, and the dynamic coordination degree is always on a coordinated development trajectory, while the static coordination degree in

the middle reaches and the lower reaches has actually dropped from coordination to basic coordination or even basic incoordination. The dynamic coordination degree of the former deviated from the coordinated development trajectory in 2017–2020, and the dynamic coordination degree of the latter deviated from the coordinated development trajectory in 2014–2020. The classification of the static coordination degree of various regions is obvious with significant spatial aggregation characteristics.

Thirdly, the opening-up degree, foreign direct investment, population growth, and urban–rural income gap are not advantageous for the coordination degree, while the marketization level, industrial structure, and human capital are advantageous for the coordination degree. The regression coefficients of the latter two are not significant. The regional regression results show that the impacts of driving factors on the coordination degree have obvious heterogeneity. Therefore, the YREB needs to change the mode of foreign trade development, promote the sustainable development of foreign trade, vigorously improve the quality of foreign direct investment, actively promote the market-oriented reform of factors, improve the marketization level, speed up the upgrade of the industrial structure, improve the human capital level and the quality of the population, narrow the urban–rural income gap, and promote the further coordination development of NU and WEC.

The results of this study can help to improve the theory of NU and enrich the research content on the theory of ecological civilization. The results provide a reference for other countries and regions similar to the YREB and can be used to improve the water ecological environment during the process of NU and promote the harmonious coexistence of humans and water.

**Author Contributions:** Conceptualization, D.K. and L.L.; methodology, W.H.; software, W.Y.; validation, D.K., L.L., W.H. and W.Y.; formal analysis, D.K. and X.L.; resources, L.L.; data curation, W.Y. and X.L.; writing—original draft preparation, D.K. and W.Y.; writing—review and editing, W.H. and X.L.; visualization, L.L.; supervision, W.H.; project administration, L.L.; funding acquisition, D.K. and L.L. All authors have read and agreed to the published version of the manuscript.

**Funding:** This research was funded by the Social Science Foundation of Jiangxi Province, grant numbers 22GL56D and 21JL08D; The Humanities and Social Sciences Foundation of Jiangxi Province, grant number JJ21212.

**Data Availability Statement:** Not applicable.

**Conflicts of Interest:** The authors declare no conflict of interest.

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
