# Peer review of "Study on the Coordination of New Urbanization and Water Ecological Civilization and Its Driving Factors: Evidence from the Yangtze River Economic Belt, China"

_land, doi:10.3390/land12061191_

Round 1
Reviewer 1 Report (Previous Reviewer 1)
The authors have improved this paper in the modified version, however, there are some minor problems that need to modify before acceptance.
1 The innovation of this paper needs to be highlighted in the abstract.
2This article has obtained some interesting findings through the models, but these findings need to be further verified from theory or actual conditions. Also, further highlight the contribution of this article., the following literature should be helpful for your research:(1) Adaptability analysis of water pollution and advanced industrial structure in Jiangsu Province, China (2) Compilation of Water Resource Balance Sheets under Unified Accounting of Water Quantity and Quality, a Case Study of Hubei Province.
3. Compared with the available literature, what are the theoretical contributions and application values of this study? It is suggested to enhance the corresponding discussions in the conclusion part
English presentation requires more refinement.
Author Response
We sincerely appreciate all valuable comments and suggestions, which helped us to improve the quality of the manuscript.

Reviewer 2 Report (Previous Reviewer 3)
This is a resubmitted paper. I reviewed the earlier version of the manuscript some time ago. I would like to thank the author for accepting some of my recommendations and making respective revisions. Nevertheless, I hate to admit that the critical concerns of mine have remained unanswered or poorly answered. Below, I provide Round 2 recommendations for the improvement of the paper.
I still think that the land-related component remains underexplored. The paper hardly addresses any issues related to land use of land-related effects of the urbanization or ecological development processes. This is a critical weakness of the manuscript submitted to Land.
The selection of variables remains unconvincing. Is not clear why particular parameters are chosen. For example, in line 181, the author simply says "the following secondary indicators are chosen" - why those particular ones, why not the alternative ones, why that particular set of variables, etc. This is a critical weakness of the entire section 2.2. A comprehensive discussion of the existing approaches is needed (the characteristics of various indicators, comparisons between the classification approaches, advantages and disadvantages of each of the parameters, etc.). The author should arrive at the establishment of the author's array of indicators by guiding a reader through the comprehensive discussion of the existing approaches, emphasizing the strengths and weaknesses of those approaches, and picking the indicators one by one. Now, this narrative is missing entirely, so the selection of the parameters is not convincing and thus debatable.
Section 2.3.1 - why is this particular model applied? The author should explain the selection and demonstrate the advantages of this particular approach to addressing the aim of the study. A critical discussion of previous uses of the approach is needed, as well as a discussion of alternative methods. The author should convince a reader that the used methodology is the most appropriate one and serves the aims of the study the best.
The discussion component has improved compared to the Round 1 review, but there is still room for improvement. More pieces of evidence of international studies are needed, the overall focus on Chinese studies unbalances the narrative and limits the scope. The findings must be discussed extensively through the lens of international literature, with a convincing and informative demonstration of the author's novelties or similarities with other studies. The international implications of the China-specific findings should be discussed and demonstrated.
The quality of the English language must be improved radically, mainly in terms of style (extensively long sentences, unconcise paragraphs, grammar)
Author Response
We sincerely appreciate all valuable comments and suggestions, which helped us to improve the quality of the manuscript.

Reviewer 3 Report (Previous Reviewer 2)

Author Response
We sincerely appreciate all valuable comments and suggestions, which helped us to improve the quality of the manuscript.

Round 2
Reviewer 2 Report (Previous Reviewer 3)
The paper has been improved substantially. I thank the author for accepting my recommendations and responding to them adequately. I consider revisions sufficient. The paper can be accepted after the level of the English language is improved. I strongly recommend the author run a comprehensive proofreading of the text by a native speaker of English or the MDPIU language service
The paper can be accepted after the level of the English language is improved. I strongly recommend the author run a comprehensive proofreading of the text by a native speaker of English or the MDPIU language service
Author Response
We sincerely appreciate all valuable comments and suggestions, which helped us to improve the quality of the manuscript. We have run a comprehensive proofreading of the text by the MDPIU language service. The article has undergone English language editing by MDPI.
This manuscript is a resubmission of an earlier submission. The following is a list of the peer review reports and author responses from that submission.
Round 1
Reviewer 1 Report
1. The innovation of this paper needs to be highlighted in the abstract.
2. The author introduced the research background of this paper too much, but they does not explain the realistic background of this research very well.
3. The literature review is not enough, the innovation of this paper and the contribution made by previous studies have not been clearly expressed.
4. The presentation of the method is very imprecise
5. Interpretation of results does not highlight important issues studied in this paper
6. The discussion section does not analyze the results of this paper, and provides enlightening thinking on related issues.
7. Compared with the available literature, what are the theoretical contributions and application values of this study? It is suggested to enhance the corresponding discussions in the conclusion part
8. English presentation requires more refinement
9. This article has obtained some interesting findings through the models, but these findings need to be further verified from theory or actual conditions. Also, further highlight the contribution of this article.
10. It is suggested that the authors add more descriptions of the reasons for choosing Yangtze River Economic Belt as a case study, the following literature should be helpful for your research, the following literature should be helpful for your research:(1) Decoupling economic growth from water consumption in the Yangtze River Economic Belt, China. (2)Coordination of the Industrial-Ecological Economy in the Yangtze River Economic Belt, China. (3) Development of multidimensional water poverty in the Yangtze River Economic Belt, China.
Reviewer 2 Report
Title: “Study on the coordination of new urbanization and water ecological civilization and its driving factors: Evidence from the Yangtze River economic belt, China”
Major comments:
Framing:
· The framing of the paper is very much centered in China. Therefore, it’s unclear how this study can inform other cases around the world.
· The paper does not follow the scientific writing framework.
· I suggest beginning the manuscript with a broad topic that applies to other parts of the world (e.g., climate change, the problems with urbanization, overpopulation, reduction of biodiversity and agricultural land).
· Once you do this, you can proceed with the introduction of the problem statement. Then identify the research gap, and the purpose of the paper. Maybe you want to offer some hypotheses or research questions.
· After this broad framing, then introduce the case study in China, and describe in detail as part of your methods.
· I strongly suggest separating results from discussion. For the discussion part, please highlight your contributions.
Clarification of concepts
· It seems that the authors consider the urbanization process as a positive land use change without acknowledging the problems associated with urban sprawl, displacement, loss of natural landscape and agricultural land. What do you mean by “remarkable progress” in the urbanization of China (l. 51, p.2). How is “progress” defined and how is it measured? In what ways has it been “remarkable”? Has it achieved its goals? What were the goals?
· It seems that the authors are very much in favor of “high-quality economic development,” without considering the implications for rural communities, or for poor people, or displaced people. And what is “high-quality” attributed to? Who are the winners in this urbanization process and who are the losers?
· I don’t think that “water scarcity is caused by pollution” (l. 56, p.2). Water quality issues can be addressed with adequate water treatment without affecting water quantity (related to scarcity).
· Please explain what you mean by a highly valued new urbanization level (l/87, p.2), or the highly valued measurement of water ecological civilization (l. 93, p.2). Valued by whom? What would “moderately valued” or “low valued” look like? Please provide examples.
· Please explain the metric “per capita water resources”. I haven’t seen the use this term to explain water scarcity. You can have the same amount of water and serve different amount of people. Is it per capita water use?
· And where does the environment fall in this metric of water resources? If China does not consider the environment as a legitimate water user, this needs to be explained in detail. After all, this is about water ecological civilization, right?
· There are some terms that need to be defined and explained, when first introduced. For example, “water ecological civilization”, “Yangtze River Economic Belt”. When were these theories developed? On what contexts? For what purposes? These are explained, but much later and not in enough detail.
· In addition, let’s introduce the readers to the Five-Year Plans. What are these plans? What are the goals of the plans? Who makes these plans? How have they improved or not the growth of China? In what ways?
· It’s hard to believe that all the provinces in Figure 1 are cities (l.148, p.3). Please differentiate between the two terms.
Figures and tables.
· Table 1. For the subtitle “indicator layer”, does it refer to “secondary indicators” explained above? Likewise, does the subtitle “Target Layer” refer to the main variables? What is “indicator efficacy” and what does the “+” and “-“ mean? And, do you mean “unit of analysis” (as opposed to “dimension”)? Finally, it would be helpful for the readers to introduce the abbreviation of the indicators here.
· It’s not clear what result led to the statement “the development of new urbanization is still insufficient” (l. 279). Likewise, how do you know that the growth rate of new urbanization in the YREB and its regions reached the maximum in 2019 (l. 281)? Please provide evidence for your statements. Or are these statements based on numbers displayed in Table 4? If so, you need to direct the readers to the table and locate the table above the statements. Same comment for the other tables and figures.
· Figure 1 needs to be situated in a larger context (where in China?) and locate the Yangtze River. In addition, since this paper is about urbanization, let’s include the extents of the cities.
Minor comments:
· I suggest introducing acronyms the first time they are being used (e.g., CPC) and henceforth, only using the acronym in the manuscript. Maybe the Yangtze River Economic Belt can become another acronym?
· The way I understand this paper, the Yangtze River Economic Belt is a case study (not a research object) (l. 133, p.3).
· The indicators (primary and secondary) need to be explained.
· There are multiple grammar and spelling errors. The manuscript would benefit from editorial support and English-language support.
Reviewer 3 Report
My recommendations on the possible reworking of the manuscript are given below.
The relevance of the study should be explained more thoroughly. The author starts with the economic belt right away, but this narrative can be unfamiliar to non-Chinese readers. The international dimension is missed almost entirely, which substantially degrades the potential contribution of the study to the literature. The author should start with explaining the relevance of studying the urbanization-environment system in the international context and then proceed with China as one of the examples. How do the new urbanization and the ecological civilization principles correlate with the international environmental efforts and the sustainable development goals? What are the exact definitions of the new urbanization and the ecological civilization and how can they be interpreted in the international context?
Lines 174-175 and the entire section 2.2: it is not clear why these particular indicators are chosen. A comprehensive discussion of the existing approaches is needed (the characteristics of various indicators, comparisons between the classification approaches, advantages and disadvantages of each of the parameters, etc.). The author should arrive at the establishing of the author's array of indicators by guiding a reader through the comprehensive discussion of the existing approaches, emphasizing the strengths and weaknesses of those approaches, and picking the indicators one by one. Now, this narrative is missing entirely, so the selection of the parameters is not convincing and thus debatable.
Section 2.3.1 - why is this particular model is applied? The author should explain the selection and demonstrate the advantages of this particular approach to addressing the aim of the study. A critical discussion of previous uses of the approach is needed, as well as the discussion of alternative methods. The author should convince a reader that the used methodology is the most appropriate one and serves the aims of the study the best.
In general, the discussion component is rather poor, which significantly reduces the potential applicability of the study and the interest to this paper. More evidences of international studies are needed - for example, the introduction exclusively cites Chinese scholars, and the overall focus on Chinese studies unbalances the narrative and limits the scope. The discussion section is missing entirely, which is certainly a weakness. The findings must be discussed extensively through the lens of previous studies (international ones as well) with a convincing and informative demonstration of the author's novelties or similarities with other studies. The international implications of the China-specific findings should be discussed and demonstrated.
The land-related component must be strengthened. Currently, the study barely addresses any issues related to land use of land-related effects of the urbanization or ecological development processes. I consider this lack of focus on land issue a radical weakness of the manuscript submitted to Land.
Figure 1 and Figure 3- the author should put the area under study on a larger map with indication of other countries, directions (north, south), and other geographical indications, so non-Chinese readers can easily understand the exact location of the territory.
The quality of the English language must be improved radically, mainly in terms of style (extensively long sentences, unconcise paragraphs, grammar).